# Cortical encoding of phonetic onsets of both attended and ignored speech in hearing impaired individuals

Sara Carta[1,2]*, Emina Aličković[3,4], Johannes Zaar[3,5], Alejandro López Valdés[2,6‡], Giovanni M. Di Liberto [1,2‡]*

**1** School of Computer Science and Statistics, ADAPT Centre, Trinity College Dublin, Dublin, Ireland, **2** Trinity College Institute of Neuroscience, Trinity College Dublin, Dublin, Ireland, **3** Eriksholm Research Centre, Oticon A/S, Snekkersten, Denmark, **4** Department of Electrical Engineering, Linköping University, Linköping, Sweden, **5** Department of Health Technology, Hearing Systems Section, Technical University of Denmark, Kgs. Lyngby, Lyngby, Denmark, **6** Global Brain Health Institute, School of Engineering, Trinity Centre for Biomedical Engineering, College Dublin, Dublin, Ireland

‡ ALV and GMDL are senior author on this work.
* cartas@tcd.ie (SC); diliberg@tcd.ie (GMDL)

**Data Availability Statement:** The consent given by participants at the outset of this study did not explicitly detail sharing of the data in any format. This limitation is in keeping with EU General Data

## Abstract

Hearing impairment alters the sound input received by the human auditory system, reducing speech comprehension in noisy multi-talker auditory scenes. Despite such difficulties, neural signals were shown to encode the attended speech envelope more reliably than the envelope of ignored sounds, reflecting the intention of listeners with hearing impairment (HI). This result raises an important question: What speech-processing stage could reflect the difficulty in attentional selection, if not envelope tracking? Here, we use scalp electroencephalography (EEG) to test the hypothesis that the neural encoding of phonological information (i.e., phonetic boundaries and phonological categories) is affected by HI. In a cocktail-party scenario, such phonological difficulty might be reflected in an overrepresentation of phonological information for both attended and ignored speech sounds, with detrimental effects on the ability to effectively focus on the speaker of interest. To investigate this question, we carried out a re-analysis of an existing dataset where EEG signals were recorded as participants with HI, fitted with hearing aids, attended to one speaker (target) while ignoring a competing speaker (masker) and spatialised multi-talker background noise. Multivariate temporal response function (TRF) analyses indicated a stronger phonological information encoding for target than masker speech streams. Follow-up analyses aimed at disentangling the encoding of phonological categories and phonetic boundaries (phoneme onsets) revealed that neural signals encoded the phoneme onsets for both target and masker streams, in contrast with previously published findings with normal hearing (NH) participants and in line with our hypothesis that speech comprehension difficulties emerge due to a robust phonological encoding of both target and masker. Finally, the neural encoding of phoneme-onsets was stronger for the masker speech, pointing to a possible neural basis for the higher distractibility experienced by individuals with HI.

Protection Regulation and is imposed by the Research Ethics Committees of the Capital Region of Denmark. These ethical restrictions prevent Eriksholm Research Centre from fully anonymising and sharing the dataset. To inquire on the possibility to access the data, please contact Claus Nielsen, Eriksholm research operations manager, at clni@eriksholm.com.

**Funding:** This work was conducted with the financial support of the William Demant Fonden (https://www.williamdemantfonden.dk/), grant 21-0628 and grant 22-0552 and of the Science Foundation Ireland Centre for Research Training in Artificial Intelligence (https://www.crt-ai.ie/), under Grant No. 18/CRT/6223. This research was supported by the Science Foundation Ireland under Grant Agreement No. 13/RC/2106_P2 at the ADAPT SFI Research Centre (https://www.sfi.ie/sfi-research-centres/adapt/) at Trinity College Dublin. ADAPT, the SFI Research Centre for AI-Driven Digital Content Technology, is funded by Science Foundation Ireland through the SFI Research Centres Programme. There was no additional external funding received for this study.

**Competing interests:** The authors have declared that no competing interests exist.

# Introduction

In multi-talker scenarios, comprehension of a selected speech stream (target) is hampered by the presence of competing speech streams (maskers). Our auditory system avails of spectro-temporal and spatial cues to segregate target and masker streams. Nonetheless, masker sounds cannot be fully ignored, as that would prevent us from switching our focus of attention at will. In fact, a wealth of neurophysiology research on auditory streams segregation indicates that neural signals encode both target and masker streams, even though those encodings can be different in terms of what speech features they reflect and cortical origins [1–4].

Research on the neurophysiology of attentional selection has shed light on this process in participants with normal hearing (NH) with both invasive and non-invasive neural recordings, indicating that the focus of attention modulates the neural processing of progressively more abstract speech properties, from the sound envelope, to phonemes, and lexical information [2,4–7]. However, there remains considerable uncertainty on how attentional selection unfolds for individuals with hearing impairment (HI). This is particularly important to understand their comprehension difficulties in noisy multi-talker scenarios, leading to high distractibility and an increased listening effort [8]. Here, we test the hypothesis that, differently from previous findings with individuals without HI (Teoh et al., 2022 [7]), the cortex of individuals with HI encodes phonological information of both target and masker speech streams when considering challenging multi-talker listening scenarios (Lunner et al., 2020 [12]).

Acoustic properties of masker speech streams, such as the sound envelope, have been shown to be reliably encoded in neural activity [9]. Recent research in young NH individuals indicated that phonological information is only encoded for the target speech instead, with no evidence for phonological encoding for the masker speech [7]. However, the impact of attentional modulation on the neural encoding of phonological information in populations with HI remains to be defined. Investigating this issue is critical to understand the attentional strategies employed by listeners with HI in naturalistic multi-talker environments [10–15].

Previous studies have presented us with a complex array of findings as to how the degree of hearing loss changes how attentional modulation influences the disparity in the neural tracking of target and masker speech streams [16,17]. While NH and HI individuals exhibited a similar impact of attentional modulation on the envelope tracking, a certain degree of envelope over-representation emerged in HI listeners [17], Further research suggested that higher-order speech processing more greatly reflect attentional focus [14,18], with increased top-down active suppression of the masker speech under more challenging listening conditions [19]. However, such results only coarsely disentangled low and higher-order processes, by solely relying on the assumption that longer latencies represent higher-order processes [20,21]–without pinpointing the exact acoustic-linguistic properties that are encoded and suppressed in individuals with HI.

Here, we investigated the neural encoding of target and masker speech streams in individuals with HI, with the hypotheses that: 1) They would exhibit stronger cortical encoding of phonological information for the target than for the masker speech, reflecting their focus of attention; and 2) the encoding of some phonological properties (phoneme onsets and acoustically-invariant phonetic features) would be robustly represented for both target and masker speech streams, potentially contributing to the listening difficulties and high distractibility of individuals with HI in multi-talker auditory scenes [22].

To test these hypotheses, we carried out a re-analysis of an existing EEG dataset [18], where scalp EEG signals were recorded from participants with HI as they performed a selective-attention task in an realistic multi-talker scene. Multivariate regression was used to estimate the Temporal Response Function (TRF) describing the linear mapping from acoustic and

phonetic features to the EEG signals [23,24]. TRF models corresponding to target and masker speech streams were studied, providing us with objective neural measures of the cortical encoding of acoustic and phonetic features for testing the hypotheses of this study. Our results complement the existing literature, offering novel insights on the impact of attentional selection on the neurophysiology of speech in populations with HI, and contributing to future hearing impairment research and hearing aid development.

## Methods

### Ethics statement

Participants gave their written informed consent to participate in this study. The study was approved by the ethics committee for the capital region of Denmark (journal number H-1-2011-033). Data collection was carried out between October 1st 2019 and January 23rd 2020.

### Participants

Thirty-four hearing-impaired individuals (10 females and 24 males, aged between 21 and 84 years, mean age = 64.2 years, standard deviation = 13.6 years) participated in this study. All participants were native Danish speakers recruited from the Eriksholm Research Centre Hearing Aid User Research Participant Database. This databased is managed by certified research audiologists and abides by the best accepted practices in diagnosis and management of hearing loss. All participants in this study cohort had previously been fitted with hearing aids and had at least four months usage experience. The cohort's hearing profile spanned from mild to moderately severe (bilateral average hearing threshold of 47.5 dBHL based on a 4-frequency threshold average at 0.5kHz, 1kHz, 2kHz and 4kHz) symmetrical (<10 dBHL in three adjacent frequencies) sensorineural hearing impairment. Hearing aids were fitted binaurally with hearing aids available in the market (Oticon Opn S 1™ mini-Receiver-in-the-ear). The Voice Align Compression rationale was used for amplification, which offers a dynamic-range compression scheme with higher compression at lower input levels, also based on each participant's hearing profile. Participants reported no history of neurological or psychiatric disorders and had normal or corrected-to-normal vision. [18]. The wide age range of this experiment is justified by the presence of two participants below 50 years of age. Excluding those younger subjects (a 21-year-old and a 45-year-old), the age range becomes a more typical 54–84 years and, when re-running the analysis after excluding them from the dataset, the statistical results did not change. Detailed information regarding bilateral air conduction hearing thresholds, age and gender of the participants can be found in S1 Table.

### Experimental design

The experiment consisted of a simulation of a free-field cocktail-party scenario, with speech stimuli simultaneously delivered to the participants from an array of six loudspeakers placed around them (at ±30˚, ±112.5˚ and ±157.5˚ azimuth relative to the participants' location). (Fig 1A). Participants were instructed to selectively listen to a particular speech stream (target), corresponding to a specific talker and spatial location in the foreground, while ignoring all other talkers (masker and background babble noise). Target and masker speech stimuli were always played from the two frontal loudspeakers (*S1* and *S2*), while the four loudspeakers in the background (*B1-B4*) simultaneously presented a 4-talker babble noise each, creating a 16-talker background noise babble. Both target and masker streams were presented at 73 dB sound pressure level (SPL), while each of the noise babbles was delivered at a level of 64 dB SPL, resulting in a +3dB signal-to-noise ratio (SNR) of the frontal audio streams relative to the background.

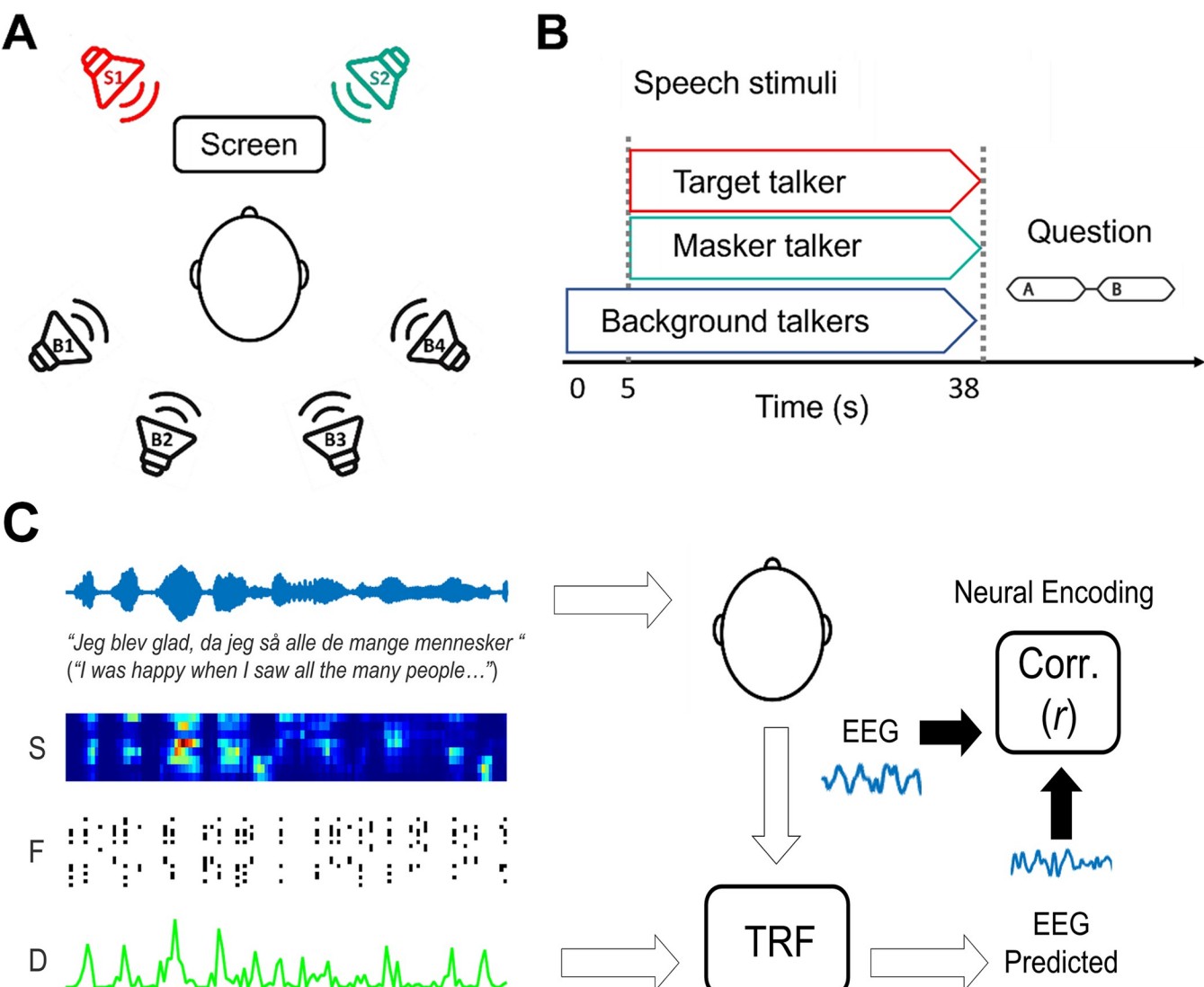

**Fig 1. Experimental design and analysis framework. (A)** Location of loudspeakers (for this particular example, S1: Target speech, S2: Masker speech, B1-B4: Babble noise) relative to the participant. **(B)** Timeline of each experimental trial. Start of the noise babble at $t_0 = 0$, followed by target and masker at $t_1 = 5s$. **(C)** Neural signals were recorded with EEG as participants listened to natural speech monologues. A lagged linear regression analysis was carried out to estimate the Temporal Response Function (TRF) describing the relationship between speech phonetic features (S—Spectrogram, F -Phonetic Features, D—Half-wave rectified Spectrogram Derivative) and low-frequency EEG (1–8 Hz).

Audio stimuli were presented at a sampling rate of 44100 Hz, delivered through a sound card (RME Hammerfall DSP multiface II, Audio AG, Haimhausen, Germany) and played through six loudspeakers Genelec 8040A (Genelec Oy, Iisalmi, Finland). EEG data was simultaneously acquired on a BioSemi ActiveTwo system with a sampling rate of 1024 Hz, from sixty-four electrodes, mounted on a standard cap according to the International 10/20 system [25]. An active and a passive electrode, CMS and DRL respectively, served as reference to all electrodes, and two electrodes were placed on the mastoids.

This study presents a novel re-analysis of a previously existing dataset from Aličković et al. (2020, 2021). Note that only two out of the four experimental blocks were re-analysed as the rest of the data was not relevant to the goals of this study. As such, the results presented here were derived based on two blocks of twenty trials each, with a listening time of eleven minutes per block.

Despite having an identical selective-attention task, the two blocks differed in the perceived loudness of the background noise, as a consequence of turning on or off a noise-reduction (NR) scheme in the participants' hearing aids. In the NR ON block [25], the NR algorithm used a beamformer with a minimum-variance distortionless response to attenuate the babble noise coming from the background [26]. In the NR OFF block, the NR scheme was deactivated and, as such, participants were completely exposed to the background noise. The NR OFF–NR ON comparison was analysed here to assess the impact of background noise on the neural encoding of target and masker speech during the task of selective attention, with the expectation that the suppression of the noise babble in the NR ON block would have improved the separability of cortical responses to target and masker streams in the foreground.

Each 20-trial block was composed of four sub-blocks of five randomized consecutive trials for each of "left male (LM)," "right male (RM)," "left female (LF)," and "right female (RF)." Before each sub-block of 5 trials, a visual cue was presented on the screen located in front of the participants, indicating the sex and location of the frontal speaker they needed to attend to.

Participants underwent a familiarisation phase, with a brief training of 4 trials, one for each of the above combinations of target speakers' sex and location of presentation. Each trial started with the presentation of the background babble noise and, after 5 seconds, the frontal target and masker speakers were added to the auditory scene (**Fig 1**). Following the end of the trial, participants were required to answer a two-alternative-forced-choice question on the target-speech content to check for their sustained attention to the task. After each block, participants could take a self-paced break before continuing with the experiment.

Individual speech stimuli consisted of short Danish newsclip monologues of neutral content (33s each). To control for differences between male and female target talkers, stimuli were all root-mean-squared (RMS) normalized to the same overall intensity. All silences were shortened to 200ms, and the long-term average spectrum of the babble noise was matched to the overall spectrum of male and female foreground talkers to avoid inconsistencies in masking.

## Stimuli feature extraction

Here, we sought to determine how specific speech properties in this competing-talker attention scenario contribute to the neural-tracking response of target and masker speakers. Therefore, the foreground speech streams were modelled according to a set of features representing their low-level acoustic and phonetic characteristics. For the former set of features, we first extracted the stimuli's spectrograms by applying a filter which partitioned the sound waveforms into eight logarithmically spaced frequency bands, from 250 Hz to 8 kHz (Hamming window, with a length of 18ms, a shift of 1ms; FFT with 1024 samples), according to Greenwood's equation [27]. We also extracted, for each frequency band, its half-wave rectified spectrogram derivative. Together, these acoustic features are referred to as '**S**.'

A second set of features was derived to capture categorical phonetic features, describing the sounds typical of the Danish language in terms of articulatory features, e.g., voicing, manner of articulation, and place of articulation. This feature-set is referred to as '**F**.' First, speech stimuli were automatically transcribed, and the accuracy of the transcription was verified by a native Danish speaker. The transcripts were then processed by NordFA, a forced phonetic aligner for Nordic languages [28,29], which identified timestamps corresponding to the start and end of each word into its constituent phonemes. The quality of the forced alignment was manually verified using Praat software on about 10% of the speech material, and with custom-made MATLAB scripts to assess that phoneme onsets corresponded to increases in the envelope, as expected in case of satisfactory phoneme alignment. Each phoneme was then represented as a binary vector with eighteen dimensions, each corresponding to one articulatory feature.

Specifically, the following eighteen features were considered: syllabic, long, consonantal, sonorant, continuant, delayed-release, approximant, nasal, labial, round, coronal, anterior, dorsal, high, low, front, back, tense.

## EEG data preprocessing

Neural data were analysed by using MATLAB software (MathWorks). Analysis code was customized based on publicly available resources (please see the CNSP initiative website: https://cnspworkshop.net). Minimal EEG processing was carried out [30]. EEG data were bandpass filtered between 1 and 8 Hz [6,31] using a zero-phase shift 4th order Butterworth filter and then downsampled from 1024 to 64 Hz. Noisy EEG channels with a variance three times greater than that of the surrounding sites were replaced by means of a spherical spline interpolation, using a library in the EEGLAB Software [32]. Subsequently, we re-referenced EEG signals to the average of the two mastoid channels. Single-subject EEG was also standardised by its overall standard deviation, preserving the ratio across electrodes.

## The Temporal Response Function (TRF) framework

We employed a system identification technique, known as the TRF framework, to estimate the relationship between neural signals and speech features, which can be conventionally approximated as a linear mapping. The TRF represents the optimal set of weights, obtained by employing regularised linear regression (Crosse et al., 2016 [23]), describing the transformation from the stimulus features to the corresponding brain responses. To avoid overfitting, leave-one-out cross-validation was applied across trials, by exploring a parameter space spanning from $10^{-6}$ to $10^4$, in search for the optimal lambda, which was selected as the regularisation value yielding the highest Pearson's correlation between predicted and observed EEG data (forward model). Please note that in the case of multivariate speech representations consisting of both categorical and continuous variables (e.g., the FS model with spectrogram, spectrogram derivative and phonetic features) a banded regression was applied, in which the lambda search allowed for a different lambda selection for each set of features, aiming at the optimal combination of lambda values. Considering that the relationship between any stimulus feature and the resulting EEG response is not instantaneous, TRFs are always estimated by taking into account multiple time-lags. In this case, we considered a time-lag window spanning the interval from -100 ms to 350 ms. Since the stimulus-EEG relationship is estimated for each electrode separately and defined by examining the stimulus effect on the EEG at multiple latencies, TRF weights are interpretable both spatially, through topographies representing the scalp locations where this relationship is stronger or weaker, and temporally, by examining how the stimulus' impact on the EEG signal evolves over time.

## Speech feature models

The TRF framework was used to determine the relationship between low-frequency EEG signals and the phonetic information in speech. For both NR-ON and NR-OFF conditions, single-subject TRF models were fit for target and masker stimuli separately.

First, forward TRF models were fit considering the phonetic features (F feature-set), yielding the F model, and a control F*sh* model, with shuffled phonetic features (shuffled F feature-set). Comparing the predictive performance of the F and the F*sh* models, allows us to isolate neural variance that can be explained by the encoding of phonological information.

Next, we performed other TRF analyses to pinpoint the specific aspects contributing to the neural encoding of phonological information, particularly focussing on the acoustically-invariant encoding of phonetic categories, and on the neural encoding of phoneme boundaries, i.e.,

phoneme onsets. To investigate the cortical representation of phonetic categories, TRFs were fit by considering low-level acoustic features only (S feature-set), and a combination of F and S (FS feature-set combination). By subtracting the EEG prediction correlations for the FS and S models we derived a metric called PhCat (also called FS-S in the literature), which quantifies EEG variance explained by phonetic features but not by the acoustic features, reflecting acoustically invariant responses. To analyse the neural encoding of phoneme onsets, which would reflect a representation of phoneme boundaries rather than their specific categorical identity, the TRF model fit was re-evaluated after a random shuffling of the phonetic features identity in the FS model, producing the F*sh*S model. Here, phonemes were shuffled while retaining the temporal information, by leaving phonemes' timestamps intact. The difference between the F*sh*S model's prediction correlations and those obtained from the S acoustics-model will represent the unique contribution of phoneme onsets. This prediction correlation gain, (F*sh*S-S), is called the PhOnset metric, because it quantifies the prediction gain due to phoneme onsets while accounting for the acoustics.

## Phonetic distance maps

To determine how attentional selection alters the sensitivity to phonetic features, EEG phoneme-distance maps (PDM; Di Liberto et al., 2015 [35]; Di Liberto et al., 2021 [6]) were compared for the target and masker TRF weights of the PhCat metric (FS-S). The rationale is to project the TRF weights onto a multi-dimensional space where the Euclidean distance between phoneme pairs corresponds to the difference in their EEG responses. In doing so, these maps allow us to determine the sensitivity of the EEG signals to specific phonetic features. PDMs were obtained as follows: First, phoneme weights were calculated as a linear combination of the weights for the corresponding phonetic features. Then, a multi-dimensional scaling analysis (MDS) was carried out to reduce the dimensionality of the TRF weights, considering all time-lags and electrodes simultaneously, while preserving the standardized Euclidean distances between the EEG signals corresponding to different phonemes. The masker PDM was then mapped onto the reference target PDM, for each NR, by isomorphic scaling via Procrustes analysis (MATLAB function *procrustes*).

Four sets of phonetic categories were selected arbitrarily to reflect the major phonetic articulatory groups, ensuring that phonemes belonging to each set would create perceptually relevant phonetic contrasts defining the phonetic features: manner of articulation, place of articulation (lips), place of articulation (tongue), and voicing. For example, when considering the place of articulation (tongue) phonetic set, it is possible to explore the relative similarity of TRF weights in response to the phonetic features: high, low, back and front. If these phonetic contrasts, related to articulatory tongue movements, are perceptually relevant and represented at the neural level, then the TRF weights in response to each of these phonetic categories should have consistently similar responses, while they would be consistently different across categories.

Please note that the symbols used to represent phonetic features are an adaptation of the IPA symbols for Nordic languages. The '?' symbol indicates the Danish phonation feature "Stød", which is a type of glottal stop [33].

## Statistical analysis

All statistical analyses directly comparing the groups were performed using repeated measures three-way ANOVAs. One-sample *t*-tests were used for post-hoc tests. Correction for multiple comparisons was applied where necessary via the Holm correction. In that case, the adjusted *p*-value was reported. Descriptive statistics for the neurophysiology results are reported as a

combination of mean and standard error (SE). All analyses were performed using the JASP software [34].

## Results

Behavioural scores indicated that participants were able to successfully perform the task, displaying a 73% correct performance on 2-choice questions regarding the target speech for the NR OFF condition, and an 84% correct performance in the NR ON condition, with a significant impact of NR scheme on behavioural accuracy [18].

### Robust phonetic-feature TRF for target but not masker speech in listeners with HI

Forward TRF models were fit to characterise the mapping between phonological information and low-frequency (1–8 Hz) EEG signals. Separate TRF models were fit for target and masker speech streams, first with a phonetic-feature model F and then with a control model, F$sh$, where the phonetic categories were shuffled, while preserving their timing. A repeated-measures three-way ANOVA was conducted to evaluate the impact of NR (OFF and ON), model (F and F$_{sh}$) and attention (target and masker) on the EEG prediction correlations. This analysis indicated a main effect of attention ($F(1,33) = 24.50$, $p = 2.14e{-}05$, $\eta p^2 = 0.43$), with larger EEG prediction correlations for the target compared to the masker speech ($r_{target} > r_{masker}$, post-hoc $t$-test: $t(33) = 4.95$, $p = 2.14e{-}05$, Cohen's $d = 0.68$); a main effect of model ($F(1,33) = 31.62$; $p = 2.93e{-}06$, $\eta p^2 = 0.49$), with the F model showing greater EEG prediction correlations than its control model, F$sh$, derived by re-running the TRF computation after shuffling the corresponding phoneme categories ($r_F > r_{Fsh}$; post-hoc $t$-test: $t(33) = 5.62$, $p = 2.93e{-}06$, Cohen's $d = 0.54$); and a significant Attention*Model interaction ($F(1,33) = 18.82$, $p = 1.28e{-}04$, $\eta p^2 = 0.36$). The interaction effect revealed that an increase in EEG prediction correlation in the F model compared to its shuffled version F$sh$ ($rF > rF sh$) emerged for the target stimulus (post-hoc $t$-test: $t(33) = 7.10$, $p = 6.98e{-}09$, Cohen's $d = 0.87$) whereas this effect did not emerge for the masker speech (post-hoc $t$-test: $t(33) = 1.77$, $p = 0.16$, Cohen's $d = 0.21$). Finally, we did not find a main effect of NR ($F(1,33) = 0.17$; $p = 0.68$, $\eta p^2 = 0.005$).

For simplicity, we only report plots of the EEG prediction correlations (averaged across all EEG channels) and single-channel TRF weights (**Fig 2B**) for the NR OFF condition. Statistical post-hoc tests showed that the F model yields significantly higher prediction scores than its shuffled F$sh$ version for the target (post-hoc $t$-test: $t(33) = 5.11$, $p = 2.96e{-}05$, Cohen's $d = 0.85$), but not for the masker speech (post-hoc $t$-test: $t(33) = 0.65$, $p = 1$, Cohen's $d = 0.15$). Furthermore, an attentional effect emerged, since EEG prediction correlation values were significantly greater for the target compared to the masker, for both the F model (post-hoc $t$-test: $t(33) = 4.67$, $p = 1.96e{-}04$, Cohen's $d = 0.1$) and the F$sh$ model (post-hoc $t$-test: $t(33) = 1.79$, $p = 1$, Cohen's $d = 0.38$); (**Fig 2A**). Similarly, for the NR ON condition, the F model's EEG prediction correlations were significantly greater than those of the control F$sh$ model in the case of the target stream (post-hoc $t$-test: $t(33) = 5.41$, $p = 9.14e{-}06$, Cohen's $d = 0.9$), while this result did not emerge for the masker (post-hoc $t$-test: $t(33) = 1.23$, $p = 1$, Cohen's $d = 0.2$); (**S1A Fig**). As for the NR OFF condition, also in the NR ON the accuracy of the EEG prediction was overall greater for the target compared to the masker speech, but for the F model only (post-hoc $t$-test: $t(33) = 4.8$, $p = 1.31e{-}04$, Cohen's $d = 1$), while in this portion of the data there was no difference between target and masker in the F$sh$ model (post-hoc $t$-test: $t(33) = 1.54$, $p = 1$, Cohen's $d = 0.33$). TRF weights for channel FCz in the NR ON condition are visualised in **S1B Fig**.

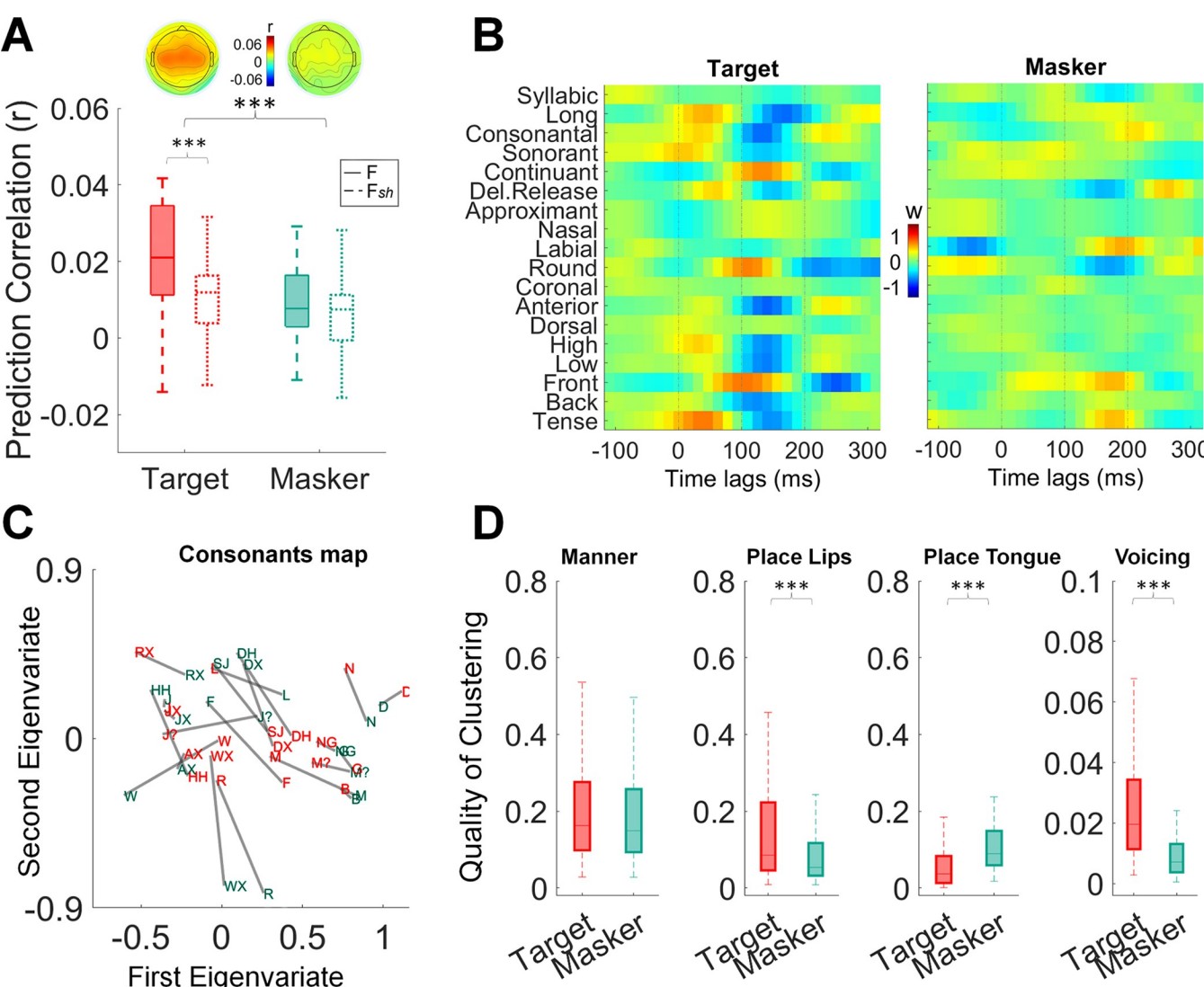

**Fig 2. Stronger cortical encoding of target acoustic-phonetic information compared to masker information in listeners with hearing impairment.** **(A)** EEG prediction correlations (Pearson's r) for phonetic features models F and shuffled control $F_{sh}$ of the target and masker speech. Boxplots show the median and inter-quartile range (IQR) of the distributions. Scalp topographies represent the distribution of prediction correlations across all channels (average across participants) for the F model. **(B)** TRF weights at channel FCz for the eighteen phonetic features for the F model. **(C)** Phoneme distance maps (PDMs) for target (red) and masker (green) speech. **(D)** EEG sensitivity to groups of phonetic features, i.e., quality of clustering of the EEG responses around relevant phonetic contrasts, for target and masker speech.

The results above indicate a robust relationship between EEG signals and phonological information of the target speech, while no such relationship is observed for the masker speech. Subsequent analyses were conducted to clarify whether attention had a more pronounced influence on the encoding of specific groups of phonetic features. To do so, PDMs reflecting the encoding of the target and masker speech were derived from the TRF weights of the target and masker F models separately (**Fig 2C**; see **Methods**; **S1C Fig** for NR ON). Calinski-Harabasz metrics were derived to evaluate the quality of clustering for both the target and masker speech. These metrics measure the degree of separation between the relevant phonetic feature clusters within the EEG-based PDMs. Larger values indicate that the organisation of PDMs aligns well with a specific group of features.

Fig 2D summarises the results of the PDM analysis indicating that, overall, the neural encoding of different phonetic feature sets is more consistently clustered in the target compared to the masker speech, while the impact of NR emerged as significant for some phonetic features sets, but not for others. These results emerged from a two-way repeated measures ANOVA–NR (ON and OFF) and attention (target and masker)—which was conducted four times, one for each phonetic feature set.

For *manner of articulation*, the analysis revealed a main effect of attention ($F(1,99) = 15.90$; $p = 1.29\text{e-}04$, $\eta p^2 = 0.14$), with a general effect of target > masker, (post-hoc *t*-test: $t(33) = 3.99$, $p = 1.29\text{e-}04$, Cohen's $d = 0.36$), while there was no effect of NR ($F(1,99) = 2.79$; $p = 0.09$, $\eta p^2 = 0.03$). In the specific case of NR OFF plotted here, there was no statistical difference between target and masker (post-hoc *t*-test: $t(33) = 1.74$, $p = 0.25$, Cohen's $d = 0.23$). For the NR ON condition (S1D Fig), the clustering of *manner*-related phonetic features was more consistent for the target speech compared to the masker (post-hoc *t*-test: $t(33) = 3.8$, $p = 9.61\text{e-}04$, Cohen's $d = 0.5$).

For *place of articulation (lips)*, a significant effect of attention emerged ($F(1,99) = 59.55$; $p = 9.65\text{e-}12$, $\eta p^2 = 0.38$), with target > masker, (post-hoc *t*-test: $t(33) = 7.72$, $p = 9.65\text{e-}12$, Cohen's $d = 0.78$). A main effect of NR was also found ($F(1,99) = 12.47$; $p = 6.31\text{e-}4$, $\eta p^2 = 0.11$), with NR ON > OFF (post-hoc *t*-test: $t(33) = 3.53$, $p = 6.31\text{e-}4$, Cohen's $d = 0.37$), together with a statistically significant interaction of attention and NR ($F(1,99) = 5.62$; $p = 0.02$, $\eta p^2 = 0.05$). For the NR OFF condition, plotted in Fig 2D, the clustering for the target speech was greater than that for the masker (post-hoc *t*-test: $t(33) = 4.10$, $p = 1.82\text{e-}4$, Cohen's $d = 0.56$). The same pattern of results emerged in the NR ON condition, plotted in S1D Fig (post-hoc *t*-test: $t(33) = 7.3$, $p = 3.57\text{e-}11$, Cohen's $d = 1$).

Regarding the *place of articulation (tongue)* features, the ANOVA revealed once again a significant main effect of attention ($F(1,99) = 45.86$; $p = 9.15\text{e-}10$, $\eta p^2 = 0.32$), this time with a more effective clustering for the masker than for the target, (post-hoc *t*-test: $t(33) = 6.77$, $p = 9.15\text{e-}10$, Cohen's $d = 0.64$). For NR OFF specifically, it can be observed that masker > target (post-hoc *t*-test: $t(33) = 3.50$, $p = 0.002$, Cohen's $d = 0.50$). Similarly, in the case of the NR ON condition (S1D Fig), the masker speech showed a more consistent clustering than the target (post-hoc *t*-test: $t(33) = 5.4$, $p = 1.19\text{e-}06$, Cohen's $d = 0.79$).

In the case of voicing, the 2-way ANOVA revealed a significant effect of attention ($F(1,99) = 221.96$; $p = 4.96\text{e-}27$, $\eta p^2 = 0.70$), with target > masker (post-hoc *t*-test: $t(33) = 14.90$, $p = 4.96\text{e-}27$, Cohen's $d = 1.46$), a significant effect of NR ($F(1,99) = 14.35$; $p = 2.61\text{e-}4$, $\eta p^2 = 0.13$), with NR ON yielding a better clustering than NR OFF (post-hoc *t*-test: $t(33) = 3.80$, $p = 2.61\text{e-}4$, Cohen's $d = 0.39$), and a significant attention*NR interaction ($F(1,99) = 48.23$; $p = 4.03\text{e-}10$, $\eta p^2 = 0.33$). For the NR OFF condition reported here, phonetic features related to voicing were more consistently clustered for the target compared to the masker speech (post-hoc *t*-test: $t(33) = 6.54$, $p = 1.03\text{e-}9$, Cohen's $d = 0.86$). Similarly, the NR ON condition (S1D **Fig**) showed a significantly greater clustering for the target compared to the masker stream (post-hoc *t*-test: $t(33) = 15.76$, $p = 7.54\text{e-}36$, Cohen's $d = 2.07$).

## Cortical encoding of both target and masker phoneme onsets in listeners with HI

The TRF F model captured phonological information, potentially reflecting both an acoustically invariant cortical encoding of phonetic features as well as responses to other correlated information, such as the acoustic spectrogram [7]. To isolate EEG signals that can be explained by phonetic feature categories but not by speech acoustics, TRF models were fit based on an FS set of features comprising the S model (spectrogram and the half-wave rectified spectrogram

derivative) and the F model (phonetic-feature categories). The gain in EEG prediction correlations due solely to phonetic features was then assessed by subtracting the FS and the S models, yielding the PhCat metric.

The PhCat metric (FS-S gain) may reflect the EEG encoding of the timing of phonetic feature categories, as well as their identity [6,35,36]. To assess if either or both these properties were encoded in the EEG signals, we also compared the FS-S predictive performance with the results of the same analysis after shuffling the phoneme identities in the transcript, which resulted in the PhOnset metric (F*sh*S-S gain). This PhOnset metric allowed us to isolate the contribution of phoneme onsets, while the PhCat metric represented the contribution of phonetic feature identity.

We compared the EEG prediction correlation gain obtained from the PhCat metric (FS-S) and the one obtained from the PhOnset metric (F*sh*S-S) using a repeated-measures three-way ANOVA with factors: NR (OFF and ON), model gain (FS-S and F*sh*S-S) and attention (target and masker). This analysis was done to dissociate the potential contribution of phoneme onsets from that of phonetic feature identity. Our analysis revealed no significant difference between the EEG prediction increase of the two model gains (PhCat and PhOnset metrics), suggesting that the contribution of phoneme onsets and phonetic feature identity is overall comparable (no main effect of model gains: $F(1,33) = 1.98$, $p = 0.17$, $\eta p^2 = 0.06$). Furthermore, we observed a significantly greater EEG prediction increase for the masker speech, compared to the target, across the two model gains (significant main effect of attention: $F(1,33) = 11.26$, $p = 0.002$, $\eta p^2 = 0.25$. $r_{masker} > r_{target}$, post-hoc $t$-test: $t(33) = 3.36$, $p = 0.002$, Cohen's $d = 0.42$). The ANOVA results did not indicate a significant main effect of NR on the EEG prediction correlations ($F(1,33) = 0.09$, $p = 0.7$, $\eta p^2 = 0.003$). An interaction between attention and model gain did reach statistical significance ($F(1,33) = 4.26$, $p = 0.047$, $\eta p^2 = 0.11$), showing that the target speech PhOnset metric yielded significantly lower EEG prediction gains compared to both the masker PhCat metric (post-hoc $t$-test: $t(33) = -3.63$, $p = 0.003$, Cohen's $d = -0.5$) and the masker PhOnset metric (post-hoc $t$-test: $t(33) = -3.94$, $p = 0.001$, Cohen's $d = -0.56$).

As in the previous section, we only report plots for the NR OFF condition, where no difference was found between the PhCat and PhOnset metrics, neither for the target speech (post-hoc $t$-test: $t(33) = 1.10$, $p = 1$, Cohen's $d = 0.15$) nor for the masker speech (post-hoc $t$-test: $t(33) = 1.05$, $p = 1$, Cohen's $d = 0.14$). The statistical analysis showed a greater predictive gain of phoneme onsets for the masker speech, compared to the target (post-hoc $t$-test: $t(33) = 3.24$, $p = 0.046$, Cohen's $d = 0.72$) (**Fig 3A and 3B**). For the NR ON condition, as shown in **S2A and S2B Fig**, we found the same pattern of results, with no significant difference between the PhCat and PhOnset metrics, neither for the target (post-hoc $t$-test: $t(33) = 2.38$, $p = 0.43$, Cohen's $d = 0.32$) nor for the masker stream (post-hoc $t$-test: $t(33) = 1.86$, $p = 1$, Cohen's $d = 0.41$).

## Discussion

Individuals with hearing impairment have difficulties in sustaining attention to the speaker of interest in multi-talker scenarios [12]. Here, we measured how attentional selection modulates the cortical encoding of phonological information in participants with HI, testing whether hearing impairment leads to a consistent cortical representation of the masker speech, resulting in high distractibility. Our results show two main patterns: First, we found that cortical responses in participants with HI relate to phonological information more strongly for the target than the masker streams, which is similar to the pattern previously measured in participants without HI [7]. Second, when isolating the different contributors to phonological information encoding, we found a significant cortical encoding of phoneme onsets of both target and masker speech, with a stronger representation of the masker.

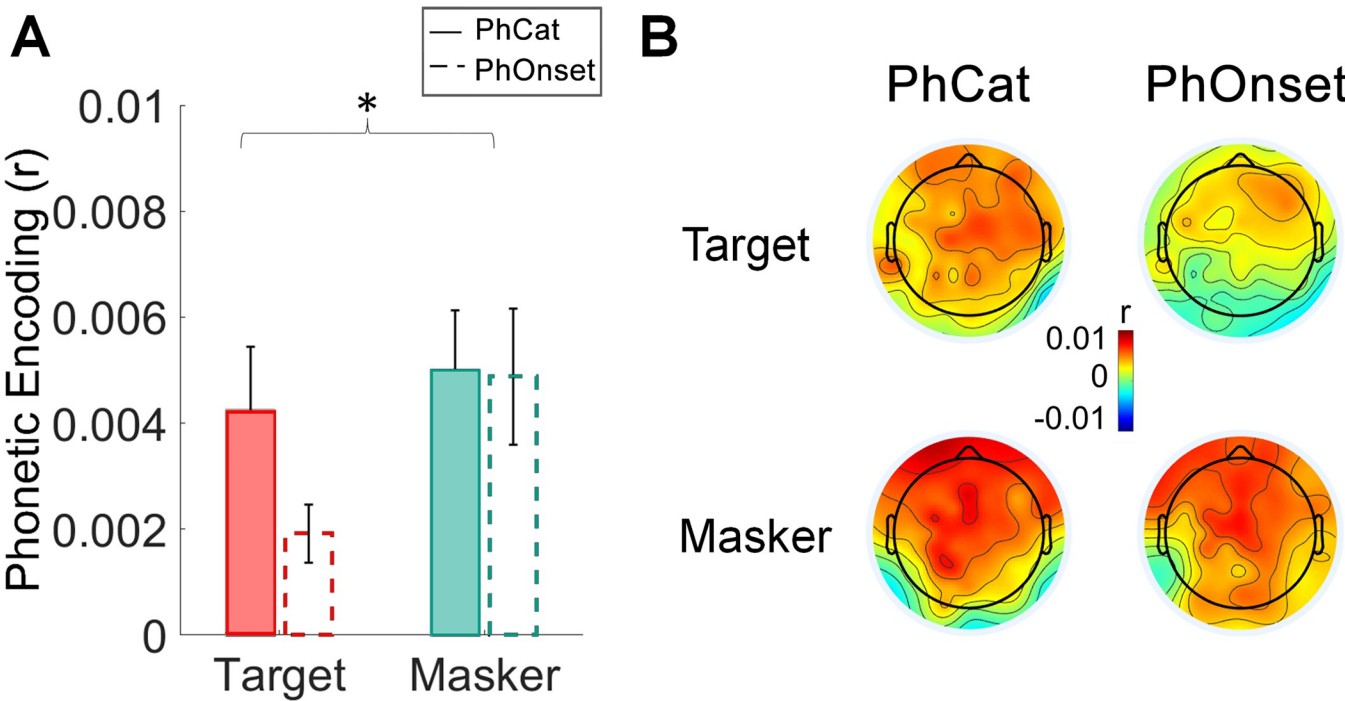

**Fig 3. Representation of phonemic onsets of the ignored speech sounds in the low-frequency EEG of participants with HI.** **(A)** EEG prediction gains obtained from the PhCat (FS-S) and PhOnset ($F_{sh}$S-S) metric, for the target and masker speech. Bars represent the increase in prediction correlations (r) averaged across all participants and electrodes. Error bars represent the SEM across participants. **(B)** Topographical distribution of the average EEG prediction correlation increases from the baseline model S, across all electrode locations.

Our findings impact the current understanding of how selective attention strategies unfold in the brain of listeners with HI. The finding of a stronger relationship between cortical responses and phonological information of the Target speech suggests that individuals with HI perform attentional filtering at the level of the phonetic encoding, segregating target and masker streams and processing phonetic information of target and masker in a different manner. However, we could not isolate the acoustically invariant component of the phonetic responses here, meaning that the effect of selective attention on the phonetic feature EEG metric likely reflects a combination of acoustically variant and invariant responses to phonetic features. While possible explanations could relate to sample size and data quality, it should be noted that this is the first study measuring phonetic feature TRFs with the Danish language, while previous studies isolating acoustically invariant phonetic-feature TRFs used the English language (and Flemish), [37]. Future work could further investigate this question by considering participants with and without HI across different languages.

Our finding that cortex of individuals with HI encodes the phoneme onsets of both target and masker speech confirms the initial hypothesis of a highly interfering representation of phoneme-level information of the masker, which could contribute to explaining the high distractibility reported in the literature. Interestingly, the cortical encoding of phoneme onsets was stronger for the masker than the target. This result suggests that the onset encoding itself changes with the specific phoneme category for the target speech, reducing the consistency across different occurrences when those categories are shuffled in F*sh*. Conversely, the stronger phoneme onset encoding for the masker suggests higher consistency across phoneme categories, as it emerged from the analysis in **Fig 2A**. Overall, these results indicate a significant encoding of phonetic distinctions for the target but not the masker, as well as a strong

representation of the masker phoneme onsets, which could underlie participants' high distractibility in the presence of HI. In the context of selective attention, the cortical tracking of the masker phoneme onsets could serve two important functions: keeping a fundamental structural representation of the to-be-ignored stimulus for an effective re-orienting of attention on the masker stimulus at later stages, and building a structural representation of the masker speech, which is useful for suppressing it.

It is worth noting that the robust cortical encoding of phoneme onsets in both Target and Masker speech streams—despite successful behavioural performance and neural attention bias at the level of overall phonological encoding—might also reflect fluctuations in participants' attention during the course of the trial. A recent electrocorticography study on NH listeners [38] found that Masker's phonetic features are neurally encoded when the Target is silent enough to allow to glimpse the Masker out of the auditory mixture, suggesting that sustained attention on the Target is not necessarily continuous and stable. The fluctuation of attentional spotlight between Target and Masker might be even more pronounced in participants with HI, particularly in the presence of changing levels of Target-Masker SNR.

It is interesting to note that the EEG-measured phoneme-level encoding was not influenced by the hearing aid noise-reduction scheme, despite increased comprehension. The NR scheme reduced the sound level of the 16-talker babble noise in the background, without affecting the relative sound level between the target and masker speech streams originating in front of the speaker. As such, while NR ON made the task easier, increasing the overall comprehension [10,18], the listener's brain still had to segregate target and masker speech streams. Based on this result, we can speculate that the impact of attentional modulation measured at the phonetic level does not reflect overall speech comprehension, but a more general cortical strategy for selective attention in participants with HI. Instead, measuring overall speech comprehension might be effective by isolating higher order processes, such as lexical predictions [39].

Previous work highlighted the key role of temporal information and phoneme onsets in the segregation of auditory streams in multi-talker scenarios. Temporal coherence was proposed as a key criterion for grouping acoustic information into separate streams [40,41]. Previous work also found acoustic onsets of naturalistic speech to be faithfully represented in the auditory cortex [42,43], guiding the segregation of target and masker streams [9]. Acoustic-onset enhancement of the target speech signal has proven to benefit sentence recognition in cochlear implant users, even in the presence of competing speech [44,45]. Furthermore, the neural encoding of acoustic onsets of the masker speech streams was suggested to serve as a template structure of the masker stream, contributing to its suppression [9]. Intriguingly, the late encoding of acoustic onsets of the masker in fronto-parietal regions was also shown to be important for attentional selection, with a stronger encoding emerging when the masker stream was presented at a higher sound level than the target stream [19]. This effect could not be studied with our experimental paradigm, which only modulated the SNR by reducing the sound level of the speech babble in the background (NR OFF vs. NR ON), not the relative sound-level of target and masker streams. Further work exploring different relative sound-levels could inform us on whether the late encoding of acoustic onsets measured in fronto-parietal areas is purely acoustic or related to phonological processing. One of the main limitations of the present study is the lack of an age-matched control group, which would have allowed us to disentangle the effects of age and hearing loss. As such, future investigations could also examine the relationship between age and the degree of hearing loss with the effects measured in the present study, further explaining what the strong representation of the masker's phonetic onsets reflects exactly.

In conclusion, this study presented novel insight into the impact of selective attention on the neural encoding of speech in listeners with hearing loss. Our results complement previous

work on attentional selection by measuring phonological neural encoding in a naturalistic multi-talker scenario. The finding that target and masker streams are segregated in the human cortex is in line with models of object-based auditory attention [46,47] and complements previous evidence on the encoding of masker speech streams [9], informing us for the first time on the neural underpinnings of this process at the phoneme level in participants with hearing impairment.

## Supporting information

**S1 Fig. Target acoustic-phonetic information is more strongly encoded than the masker's information in listeners with hearing impairment when background noise is removed in the NR ON condition. (A**) EEG prediction correlations (Pearson's r) for phonetic features models F and shuffled control $F_{sh}$ of the target and masker speech. Boxplots represent the median and inter-quartile range (IQR) of the distributions. Scalp topographies represent the distribution of prediction correlations across all channels (average across participants) for the F model. Error bars indicate the SEM across participants. (**B**) TRF weights at channel FCz for the eighteen phonetic features for the F model. (**C**) Phoneme distance maps (PDMs) for target (red) and masker (green) speech. (**D**) EEG sensitivity to groups of phonetic features, i.e., quality of clustering of the EEG responses around relevant phonetic contrasts, for target and masker speech.
(TIF)

**S2 Fig. Representation of phonemic onsets of the ignored speech sounds in the low-frequency EEG of participants with hearing impairment when background babble noise is suppressed in the NR ON condition. (A**) EEG prediction gains obtained from the PhCat (FS-S) and PhOnset ($F_{sh}$S-S) metric, for the target and masker speech. Bars represent the increase in prediction correlations (r) averaged across all participants and electrodes. Error bars represent the SEM across participants. (**B**) Topographical distribution of the average EEG prediction correlation increases from the baseline model S, across all electrode locations.
(TIF)

**S1 Table. Participants' information and bilateral air conduction hearing thresholds.**
(TIF)

## Acknowledgments

We thank Emanuele Schiaffino for his graphical help on the figures.

## Author Contributions

**Conceptualization:** Emina Aličković, Johannes Zaar, Alejandro López Valdés, Giovanni M. Di Liberto.

**Data curation:** Sara Carta.

**Formal analysis:** Sara Carta.

**Supervision:** Emina Aličković, Johannes Zaar, Alejandro López Valdés, Giovanni M. Di Liberto.

**Writing – original draft:** Sara Carta, Giovanni M. Di Liberto.

**Writing – review & editing:** Emina Aličković, Johannes Zaar, Alejandro López Valdés.

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
