## [Decision Letter · Decision Letter 0]

8 Jan 2024

PONE-D-23-27730Cortical encoding of phonetic onsets of both attended and ignored speech in hearing impaired individualsPLOS ONE

Dear Dr. Di Liberto,

Thank you for submitting your manuscript to PLOS ONE. After careful consideration, we feel that it has merit but does not fully meet PLOS ONE’s publication criteria as it currently stands. Therefore, we invite you to submit a revised version of the manuscript that addresses the points raised during the review process.

We look forward to receiving your revised manuscript.

Kind regards,

Caicai Zhang

Academic Editor

PLOS ONE

Journal Requirements:

“This work was conducted with the financial support of the William Demant Fonden (https://www.williamdemantfonden.dk/), grant 21-0628 and grant 22-0552 and of the Science Foundation Ireland Centre for Research Training in Artificial Intelligence (https://www.crt-ai.ie/), under Grant No. 18/CRT/6223. This research was supported by the Science Foundation Ireland under Grant Agreement No. 13/RC/2106_P2 at the ADAPT SFI Research Centre (https://www.sfi.ie/sfi-research-centres/adapt/) at Trinity College Dublin. ADAPT, the SFI Research Centre for AI-Driven Digital Content Technology, is funded by Science Foundation Ireland through the SFI Research Centres Programme.”

3. We noted in your submission details that a portion of your manuscript may have been presented or published elsewhere. [A different analysis of the same dataset was published as part of another study by Alickovic et al. (2021) in Frontiers in Neuroscience (https://doi.org/10.3389/fnins.2021.636060).] Please clarify whether this [conference proceeding or publication] was peer-reviewed and formally published. If this work was previously peer-reviewed and published, in the cover letter please provide the reason that this work does not constitute dual publication and should be included in the current manuscript.

4. Please update your submission to use the PLOS LaTeX template. The template and more information on our requirements for LaTeX submissions can be found at http://journals.plos.org/plosone/s/latex.

“This work was conducted with the financial support of the William Demant Foundation, grant  21-0628 and grant 22-0552 and of the Science Foundation Ireland Centre for Research Training  in Artificial Intelligence, under Grant No. 18/CRT/6223. This research was supported by the Science Foundation Ireland under Grant Agreement No. 13/RC/2106_P2 at the ADAPT SFI Research Centre at Trinity College Dublin. ADAPT, the SFI Research Centre for AI-Driven Digital Content Technology”

“This work was conducted with the financial support of the William Demant Foundation, grant 37 21-0628 and grant 22-0552 and of the Science Foundation Ireland Centre for Research Training 38 in Artificial Intelligence, under Grant No. 18/CRT/6223. This research was supported by the 39 Science Foundation Ireland under Grant Agreement No. 13/RC/2106_P2 at the ADAPT SFI 40 Research Centre at Trinity College Dublin. ADAPT, the SFI Research Centre for AI-Driven 41 Digital Content Technology”

Reviewers' comments:

Reviewer's Responses to Questions

**Comments to the Author**

1. Is the manuscript technically sound, and do the data support the conclusions?

Reviewer #1: Partly

Reviewer #2: Partly

2. Has the statistical analysis been performed appropriately and rigorously? 

Reviewer #1: Yes

Reviewer #2: Yes

3. Have the authors made all data underlying the findings in their manuscript fully available?

Reviewer #1: No

Reviewer #2: No

4. Is the manuscript presented in an intelligible fashion and written in standard English?

Reviewer #1: Yes

Reviewer #2: Yes

5. Review Comments to the Author

Reviewer #1: The article is very well written and contains relevant references to the topic discussed. However, there are some points that can be improved. Therefore, I present my considerations below.

Methods: In the ethics statement session, the information presented in lines 128 and 129 should be present in the participants session. Furthermore, there is a major flaw at this point, as the author reports that 34 individuals were evaluated but describes only 10 female individuals. Where is the information on the remaining 24 individuals? Are they male? How old is it? What is the mean age? What is the standard deviation? Is there a control group? Was the article written exclusively by individuals with hearing loss?

On the other hand, the information contained in "Data Availability" lines 301 to 309 should be detailed in ethics statement session.

Another point I would like to know is why a study that was completed at the beginning of 2020 was only submitted for consideration at the end of 2023?

Another extremely critical point in this article is in lines 137 and 138 "for furtheer details on their hearing profile, hearing aid fitting, and signal processing see ALickovic et al, 2021". Readers need access to this data in this article. It makes no sense to direct the reader to another article. Therefore, review this sentence and present all the data in this publication. It should be noted that the absence of this information makes it impossible for the reader to visualize the data and provide a correct understanding.

In the experimental design session, the authors presented very important data regarding signal processing as well as the way sound is presented. The figures are clear and well explanatory, however, information about the EEG was scarce.

I recommend that the authors provide clearer information about the EEG data, mentioning filter, sweeps and, for example, a figure with the 64 electrodes positioned. Since, the author describes that he used the international 10/20 system, but does not show or describe the positioned points (example: FPZ, FZ, OZ among others). Authors must remember that the article must serve as a basis for new research and, as the article presents itself, it does not allow for this possibility.

In the results, I did not see information about the homogeneity of the groups in relation to gender, age and hearing loss, for example. I would like to have understood the degree of hearing loss of the individuals. Did everyone have the same degree? Where's the information?

Due to the lack of information mentioned above, the discussion also does not lead the reader to understand the characteristics of hearing loss. It is essential that there is a rearrangement in the article so that the results and discussions can really provide objective and direct information. To do so, we need to understand who the individuals who were evaluated are. The article has an important methodological flaw that needs to be remedied.

Another question that intrigues would be the existence of a control group in this study. Could it be that the responses presented in individuals with hearing loss would not be similar in hearing individuals, since the competitive sound actually causes difficulties in understanding and understanding. Therefore, it is vital that there is a control group that can confirm the authors' findings.

Therefore, I recommend that the article be reorganized to comply with good methodological practices. With reformulations, adjustments to the results, discussion and conclusion are necessary.

Reviewer #2: This paper presents an experimental study on cortical encoding of phonetic onsets of both attended and unattended speech stimuli in a multi-talker condition for hearing-impaired listeners. The idea and conclusions are quite well summarised, and this work is appropriate to be published in this journal. However, there are some issues that have to addressed before publication.

1. Some sections of the paper must be presented in a more concise way, e.g., abstract, introduction and discussion. The authors should not present so many open questions in the paper, as they would for sure confuse the reader which is the focus or major question that you are going to answer. These contents should be organised in a better way.

2. There are some writing mistakes in the current form, e.g., " it was only recently that neurophysiology studies could analyse the cortical encoding...", where a verb might be missing. There are many similar problems in the paper. Also, please unify the usage of British or American English spelling, e.g., regularise or regularize. On page 18, please unify the font size. Please go throughout the manuscript and correct similar problems before resubmission.

3. It was found in experiments that cortical responses in participants with HI relate to phonetic features more strongly for the target than the masker streams, but isolating the different contributors to phonetic feature encoding results in a significant cortical encoding of phoneme onsets of both target and masker speech with a stronger representation of the masker, which is also similarly emphasised in the discussion section. Could you please explain more on this point? because these two observations seem contradictory to each other. Can phoneme onset be regarded as one of phonetic features?

4. Some interesting conclusions are drawn in the paper, however I cannot imagine the relation to practical applications, e.g., hearing aids. For hearing-aid users, we would like to enhance the target speech, suppress the competing speaker and preserve the spatial cues of all directional sources, see the recommended references.

5. In multi-talker conditions, the most important task would be target speaker extraction, if EEG is more useful than direction-of-arrival (DOA) to help speech separation?

6. In experiments, a noise reduction algorithm was utilised to see the impact of background noises, where speech distortion is inevitable. Can you quantify the speech distortion level or speech intelligibility?

7. Recommended references:

[A] J Zhang, QT Xu, QS Zhu, ZH Ling, BASEN: Time-Domain Brain-Assisted Speech Enhancement Network with Convolutional Cross Attention in Multi-talker Conditions, in Proc. ISCA Interspeech, 2023.

[B] J Zhang, C Li, Quantization-aware binaural MWF based noise reduction incorporating external wireless devices

IEEE/ACM Transactions on Audio, Speech, and Language Processing 29, 3118-3131, 2021.

[C] J Zhang, G Zhang, A parametric unconstrained beamformer based binaural noise reduction for assistive hearing

IEEE/ACM Transactions on Audio, Speech, and Language Processing 30, 292-304, 2022.

6. PLOS authors have the option to publish the peer review history of their article (what does this mean?). If published, this will include your full peer review and any attached files.

Reviewer #1: No

Reviewer #2: No

---

## [Author Response · Author response to Decision Letter 0]

27 Feb 2024

We thank the reviewers and the editor for their work on this manuscript. The comments on the experimental setup and participant information led us to further reflect on the clarity and reproducibility of the procedure, leading to an improved version of this manuscript. In our revised manuscript, we have also sought to address the concerns on the intelligibility of our text, improving its grammatical consistency. Furthermore, the comments on our experimental results and the applicability of our study have led us to interesting reflections on our work. 

Reviewer #1

The article is very well written and contains relevant references to the topic discussed. However, there are some points that can be improved. Therefore, I present my considerations below.

1. R. Methods: In the ethics statement session, the information presented in lines 128 and 129 should be present in the participants session. Furthermore, there is a major flaw at this point, as the author reports that 34 individuals were evaluated but describes only 10 female individuals. Where is the information on the remaining 24 individuals? Are they male? How old is it? What is the mean age? What is the standard deviation? Is there a control group? Was the article written exclusively by individuals with hearing loss?

A. We have reorganised that part as suggested by the reviewer, so that the statement on the participants is in the appropriate section. We have clarified the biological sex of the entire sample. Regarding the last two questions, we further clarified that the study only considered a cohort of participants with hearing loss. 

2. R. On the other hand, the information contained in "Data Availability" lines 301 to 309 should be detailed in ethics statement session.

A. We moved the “Data Availability” paragraph below the “Ethics Statement”, so that it is clear that they provide complementary information and re-organised the text so that all information appears in the appropriate sub-sections.

3. R. Another point I would like to know is why a study that was completed at the beginning of 2020 was only submitted for consideration at the end of 2023?

A. The EEG dataset used for this study was collected in 2019 and 2020, as part of another study (see Alickovic et al. 2021). The dataset was then re-analysed in the present study to answer a different research question. 

4. R. Another extremely critical point in this article is in lines 137 and 138 "for further details on their hearing profile, hearing aid fitting, and signal processing see Alickovic et al, 2021". Readers need access to this data in this article. It makes no sense to direct the reader to another article. Therefore, review this sentence and present all the data in this publication. It should be noted that the absence of this information makes it impossible for the reader to visualize the data and provide a correct understanding.

A. We agree with the Reviewer and have added that information to the Participants section, including details on the participants’ hearing profile and on the hearing aid fitting schemes. 

5. R. In the experimental design session, the authors presented very important data regarding signal processing as well as the way sound is presented. The figures are clear and well explanatory, however, information about the EEG was scarce.

I recommend that the authors provide clearer information about the EEG data, mentioning filter, sweeps and, for example, a figure with the 64 electrodes positioned. Since, the author describes that they used the international 10/20 system but does not show or describe the positioned points (example: FPZ, FZ, OZ among others). Authors must remember that the article must serve as a basis for new research and, as the article presents itself, it does not allow for this possibility.

A. We agree with the reviewer that ensuring replicability is extremely important. As such, we carefully verified that all the necessary details are available in the paper. Regarding the electrode positioning, we better specified the details of the setup, and we added a reference for further information on the International 10/20 system. Regarding the preprocessing, all the necessary information is included in the text (please note that we specify the filter type, frequency cut-offs, and order). For clarity, we have added a reference to the specific guidelines for minimal preprocessing we followed. Thank you for this comment that pushed us to double-check this potential issue.

6. R. In the results, I did not see information about the homogeneity of the groups in relation to gender, age, and hearing loss, for example. I would like to have understood the degree of hearing loss of the individuals. Did everyone have the same degree? Where's the information?

Due to the lack of information mentioned above, the discussion also does not lead the reader to understand the characteristics of hearing loss. It is essential that there is a rearrangement in the article so that the results and discussions can really provide objective and direct information. To do so, we need to understand who the individuals who were evaluated are. The article has an important methodological flaw that needs to be remedied.

A. The information about participants’ hearing profile has been further detailed in the Participants’ section (see our reply to point 4). As we specified in the manuscript, the cohort of participants was homogenous in that they all had a mild-to-moderately severe sensorineural hearing loss, with an average 4-frequency PTA of 47.5 dB HL. Mean age (with standard deviation) and biological sex are stated at the start of the Participants’ section. As part of this study, we did not look into the correlation between levels of hearing loss and phonetic encoding metrics, and our results do not include these finer-grained distinctions. It would surely be interesting to use the single-subject hearing information in future studies, to better evaluate the impact of various degrees of hearing loss on the neural encoding of linguistic information. 

7. R. Another question that intrigues would be the existence of a control group in this study. Could it be that the responses presented in individuals with hearing loss would not be similar in hearing individuals since the competitive sound actually causes difficulties in listening and understanding. Therefore, it is vital that there is a control group that can confirm the authors' findings. Therefore, I recommend that the article be reorganized to comply with good methodological practices. With reformulations, adjustments to the results, discussion and conclusion are necessary.

A. We agree with the Reviewer that a control group in this study would allow us to have an immediate and fair comparison across hearing abilities, and it would be especially useful to disentangle between the effects of age (since the mean age was 64.2 years) and hearing loss. While we agree on the importance of such additional data, we think our results are valuable nonetheless in uncovering the neural underpinnings of multi-talker listening specific to individuals with hearing impairment. This result proved to be different from what has previously been shown in normal-hearing listeners (see, for instance, Teoh et al. 2022), in similar experimental tasks. In fact, our results different angle into acoustic-phonetic cortical mechanisms in hearing-impaired listeners, which we think will spark more work in this direction.

Reviewer #2

This paper presents an experimental study on cortical encoding of phonetic onsets of both attended and unattended speech stimuli in a multi-talker condition for hearing-impaired listeners. The idea and conclusions are quite well summarised, and this work is appropriate to be published in this journal. However, there are some issues that have to addressed before publication.

1. R. Some sections of the paper must be presented in a more concise way, e.g., abstract, introduction and discussion. The authors should not present so many open questions in the paper, as they would for sure confuse the reader which is the focus or major question that you are going to answer. These contents should be organised in a better way.

A. The text has been revised as suggested by the reviewer. The Introduction and Methods sections are the ones where those issues were primarily found and fixed. 

2. R. There are some writing mistakes in the current form, e.g., " it was only recently that neurophysiology studies could analyse the cortical encoding...", where a verb might be missing. There are many similar problems in the paper. Also, please unify the usage of British or American English spelling, e.g., regularise or regularize. On page 18, please unify the font size. Please go throughout the manuscript and correct similar problems before resubmission.

A. The text has been carefully inspected and corrected. Thank you for highlighting the problem.

3. R. It was found in experiments that cortical responses in participants with HI relate to phonetic features more strongly for the target than the masker streams but isolating the different contributors to phonetic feature encoding results in a significant cortical encoding of phoneme onsets of both target and masker speech with a stronger representation of the masker, which is also similarly emphasised in the discussion section. Could you please explain more on this point? because these two observations seem contradictory to each other. Can phoneme onset be regarded as one of phonetic features? 

A. This is a crucial point for the study, and we agree with the Reviewer that the apparent contradiction deserves more explanation. 

Phonetic feature encoding and phoneme onset encoding are two separate processes, that are investigated through different metrics: the first one, in Figure 2A, isolates the neural encoding of generic acoustic-phonetic information, by subtracting a shuffled phonetic feature model from a true phonetic feature model. As such, this metric excludes the contribution of phoneme onsets, and only retains acoustic-phonetic features. On the other hand, the second metric, in Figure 3A, isolates the brain encoding of phoneme onsets, through a model that explicitly considers the contribution of acoustic features, highlighting the gain that is obtained from phoneme onsets only. This point has been made clearer in the Discussion section (Line 513 and following).

4. R. Some interesting conclusions are drawn in the paper; however, I cannot imagine the relation to practical applications, e.g., hearing aids. For hearing-aid users, we would like to enhance the target speech, suppress the competing speaker, and preserve the spatial cues of all directional sources, see the recommended references.

A. Thank you for raising this interesting point. Our study is focused on characterising the struggle that people with hearing impairment face in daily listening scenarios, with the goal to develop objective neural metrics which could be useful for clinical applications. Our findings could potentially inform hearing-aid design, if properly supported by future research. For instance, based on our results, it appears that the acoustics of Target and Masker streams is effectively segregated in the EEG responses, but an overlap in the cortical encoding emerges at the level of phonetic models, potentially indicating a source of distraction for hearing-impaired listeners. This finding could be useful to devise systems that enhance somehow the phonetic distinctions between competing speakers, for example by selectively dampening particularly distracting phoneme onsets of the unattended speech. Further research would be needed to understand what makes a phoneme onset particularly distracting, but our result represents a first step in this direction. 

5. R. In multi-talker conditions, the most important task would be target speaker extraction, is EEG is more useful than direction-of-arrival (DOA) to help speech separation?

A. While DOA (and, for example, eye-tracking – see video here https://videos.files.wordpress.com/4LIJR7oT/wp6_d6.4_video03_dvd.mp4) is methodologically much simpler than using EEG, in real life scenarios we don’t always pay attention to what is in front of us or what we are looking at (e.g., listening to a backseat passenger while driving). We don’t see EEG solutions as something to be employed on their own, but as a complementary system that would be part of a wider range of metrics (including DOA). It should be noted though that decoding attention was not the goal of this particular study, as we were interested in how attention shapes the neural processing of phonetic information. 

6. R. In experiments, a noise reduction algorithm was utilised to see the impact of background noises, where speech distortion is inevitable. Can you quantify the speech distortion level or speech intelligibility?

A. We do not have metrics to quantify speech distortion as part of this study. Regarding intelligibility, after each trial, participants were required to answer a behavioural question on the content of the attended speech, which mostly served as a way to keep them engaged in the task and attentive. A difference between Noise Reduction Off and On emerged at the behavioural level, with more questions answered correctly when the background noise was suppressed. However, as we do not have metrics of speech distortion, we cannot claim that the improved behavioural performance directly depends on distortion. 

7. R. Recommended references:

[A] J Zhang, QT Xu, QS Zhu, ZH Ling, BASEN: Time-Domain Brain-Assisted Speech Enhancement Network with Convolutional Cross Attention in Multi-talker Conditions, in Proc. ISCA Interspeech, 2023.

[B] J Zhang, C Li, Quantization-aware binaural MWF based noise reduction incorporating external wireless devices

IEEE/ACM Transactions on Audio, Speech, and Language Processing 29, 3118-3131, 2021.

[C] J Zhang, G Zhang, A parametric unconstrained beamformer based binaural noise reduction for assistive hearing

IEEE/ACM Transactions on Audio, Speech, and Language Processing 30, 292-304, 2022.

A. We thank the Reviewer for these interesting suggestions. We will keep in mind these references for our future studies on attention decoding, while they appeared less relevant to this specific study instead.

---

## [Decision Letter · Decision Letter 1]

8 Apr 2024

PONE-D-23-27730R1Cortical encoding of phonetic onsets of both attended and ignored speech in hearing impaired individualsPLOS ONE

Dear Dr. Di Liberto,

Thank you for submitting your manuscript to PLOS ONE. After careful consideration, we feel that it has merit but does not fully meet PLOS ONE’s publication criteria as it currently stands. Therefore, we invite you to submit a revised version of the manuscript that addresses the points raised during the review process.

Please see my additional comments below.

We look forward to receiving your revised manuscript.

Kind regards,

Caicai Zhang

Academic Editor

PLOS ONE

Additional Editor Comments:

Because one of the original reviewers indicated unavailability, an additional reviewer was solicited. Now three reviews were returned, and they gave contradictory recommendations. Thus I have examined the revised manuscript more carefully and also re-read the authors' responses to the reviews. I think I agree with all the reviewers that this manuscript adopts a novel approach (TRF) to examine cortical tracking of attended and ignored speech in individuals with hearing impairment, and has the potential to make a significant contribution to the field. Being a first attempt in this direction, this study also has limitations (e.g., large range of age differences, lack of a control group) that should be properly addressed and acknowledged. Thus I'd like to recommend the following action from the authors, apart from addressing the specific comments from the reviewers:

1. I checked Alickovic et al. (2021), and couldn't find detailed report of the participant information. Thus I'd suggest the authors add a table in the current manuscript (or in supplementary materials), detailing the age, gender and hearing loss level for each of the 34 participants.

2. Age difference is indeed a concern, as hearing and auditory processing systems change dramatically over age. As the participants aged between 21 and 84 years, it is questionable whether the findings of the current study are primarily driven by a particular age group (e.g., old adults), or whether it apply universally to all ages. If the authors have had more participants, they could try to divide the participants into age groups, but that's not realistic here. I suggest the authors include age as a covariate into the statistical analyses (e.g., three-way ANOVAs) and check if age dramatically changes the results. Alternatively, if the participants are mainly older adults, perhaps the authors could conduct analysis on a subset of the data by excluding young adults, but the sample size will become smaller. Please also comment on how age difference may have impacted the results and the interpretation, as a limitation (apart from the sentence on line 500-502).

3. I think it'll be a good idea to acknowledge the lack of control group as another limitation of the current study.

In addition, I have a comment about the use of phonological processing and phonetic features.

In the abstract, the authors proposed a hypothesis of phonological encoding difficulty. However, this hypothesis was not explained further in the introduction, especially what measures would indicate phonological and non-phonological encoding, and the specific predictions about these measures. Also, the authors did not return to this hypothesis in the discussion, to discuss whether the results confirmed this hypothesis or not, and if so, whether fully or partly, etc. Throughout a large part of the manuscript, the authors used a different term "phonetic features" and seemed to use it against acoustics. I wonder what's the relationship between "phonetic features" and phonological encoding. It is true that phonology and phonetics are sometimes used interchangeably in the literature, but to many people they're distinct (e.g., phonology referring to abstract phonemes, whereas phonetics includes variations across speakers, phonetic contexts, etc.). It will be helpful if the authors could elaborate on the relationship between these terminologies if they're intended to be different, or, if they were intended to mean the same thing, then use one terminology consistently across the manuscript. I also have some issue with the conclusion that the results confirmed the hypothesis (again this was only briefly said in the abstract and not carefully discussed in the discussion). To many people, phonological encoding means abstract, acoustically invariant phonemes. However, following this definition, the results did not provide evidence for the encoding of acoustically invariant features in the target and masker speech streams. What the results showed was something very specific, the encoding of phoneme onsets. Thus I don't think it's appropriate to simply conclude that the results confirmed the hypothesis.

Reviewers' comments:

Reviewer's Responses to Questions

**Comments to the Author**

1. If the authors have adequately addressed your comments raised in a previous round of review and you feel that this manuscript is now acceptable for publication, you may indicate that here to bypass the “Comments to the Author” section, enter your conflict of interest statement in the “Confidential to Editor” section, and submit your "Accept" recommendation.

Reviewer #1: All comments have been addressed

Reviewer #2: All comments have been addressed

Reviewer #3: (No Response)

2. Is the manuscript technically sound, and do the data support the conclusions?

Reviewer #1: Partly

Reviewer #2: Partly

Reviewer #3: Yes

3. Has the statistical analysis been performed appropriately and rigorously? 

Reviewer #1: Yes

Reviewer #2: Yes

Reviewer #3: Yes

4. Have the authors made all data underlying the findings in their manuscript fully available?

Reviewer #1: No

Reviewer #2: No

Reviewer #3: Yes

5. Is the manuscript presented in an intelligible fashion and written in standard English?

Reviewer #1: Yes

Reviewer #2: Yes

Reviewer #3: Yes

6. Review Comments to the Author

Reviewer #1: Dear authors,

Thank you for taking the time to thoroughly read the manuscript.

As was already said, the topic is significant, the article is well-written, however there are still unanswered questions.

Although there was a legitimate worry about the way the hidden material was presented, the study cannot be published in its current form due to significant methodological issues.

the wide age span that includes everything from young adults to the elderly. This is a crucial factor because it is well known that the ability to perceive speech declines with age, which is why the volunteers were not well chosen.

Another significant factor influencing how speech sounds are perceived is the difference between the sexes. A quick report on the information from the hearing assessments is given. This data is essential to comprehending the characteristics of the participants and the rationale for the conclusions.

Without concentrating on participant data, the article's emphasis is in providing a detailed presentation of the technology. Nonetheless, the article entails assessing and analyzing the responses provided by the participants. which the writers mostly disregarded.

I heartily advise that a fresh investigation be conducted, but this time it should consider the research participants.

Reviewer #2: (No Response)

Reviewer #3: This manuscript addresses a significant topic, employing temporal response function (TRF) analysis to explore cortical responses to phonological features within sentences among individuals with hearing impairments. It offers novel insights into the auditory attentional selection mechanisms related to hearing impairments.

1. The description of the results in the Abstract is somewhat unclear: “Multivariate temporal response function analyses indicated a stronger phonetic-feature encoding for target than masker speech streams. Interestingly, robust EEG encoding of phoneme onsets emerged for both target and masker streams, in contrast with previously published findings with normal hearing participants (NH) and in line with our hypothesis that speech comprehension difficulties emerge due to a robust phonological encoding of both target and masker. Finally, the neural encoding on phoneme-onsets encoding was stronger for the masker speech, pointing to a possible neural basis for the higher distractibility experienced by individuals with HI.”

These sentences introduce the results from three minor perspectives, but it's unclear whether phonetic features, phonological features, and phoneme onsets are independent. At least, I believe that phoneme onsets are a type of phonological feature, so the Abstract's depiction of the results seems confusing. In addition, it would be better to add some logical expressions instead of merely using terms like "Interestingly" and "Finally" to relate the results.

2. Organizing the Introduction into paragraphs would make it clearer and more logically structured for the reader.

3. “All participants were native Danish speakers and had mild-to-moderately severe symmetrical sensorineural hearing impairment, with an average 4-frequency Pure Tone Average threshold of 47.5 dB hearing level.”

It would be beneficial to provide information on the degree of hearing loss to see if there are differences among individuals with varying levels of hearing impairment. At least, authors should provide the standard deviation of 4-frequency Pure Tone Average threshold of 47.5 dB hearing level.

4. Why was only 10% of the corpus manually checked? What was the focus of this manual check, and how can you ensure the remaining 90% of the corpus is problem-free and meets the requirements?

5. How were the features shuffled (from F to Fs), and how were the phoneme onsets extracted? Please add detailed information about signal processing.

6. What does PhOnset mean, and what is the definition of FshS?

7. Figure 2c's resolution is not high enough, making the image somewhat blurry. Could you explain in detail the value of the phonetic distance feature in the article? Should the article also discuss the brain processing mechanisms of phonetic distances? Additionally, why are the points in the consonant map inconsistent between Fig 2.c. and Fig S2c? What does the "?" in the consonant map represent?

8. The author only included the results of the NR ON condition in the supporting information and did not calculate or discuss in detail the differences in the brain's decoding schemes between NR ON and OFF conditions. I believe this discussion is meaningful and necessary. Why not discuss the TRF encoding response under the conditions of the noise reduction algorithm being turned on and off in more detail?

9. Furthermore, the author should provide behavioral results (such as the correct rate of multiple-choice questions in the experiment) to prove that the participants were conducting the experiment according to the instructions. If the experimental subjects were focused on a mixed speech stream or the wrong speech stream, it does not exclude the possibility of obtaining the results reported in this paper (both target and ignore speech streams have significant cortical tracking effects, and the phoneme-onset response to the ignore speech stream is stronger). Therefore, the experimental conclusions of this paper have not been fully substantiated. The author could also provide more EEG analysis results to prove that the participants indeed focused on the correct speech stream as per the experimental setup.

7. PLOS authors have the option to publish the peer review history of their article (what does this mean?). If published, this will include your full peer review and any attached files.

Reviewer #1: No

Reviewer #2: No

Reviewer #3: No

---

## [Author Response · Author response to Decision Letter 1]

15 May 2024

Editor Comments

Because one of the original reviewers indicated unavailability, an additional reviewer was solicited. Now three reviews were returned, and they gave contradictory recommendations. Thus, I have examined the revised manuscript more carefully and also re-read the authors' responses to the reviews. I think I agree with all the reviewers that this manuscript adopts a novel approach (TRF) to examine cortical tracking of attended and ignored speech in individuals with hearing impairment and has the potential to make a significant contribution to the field. Being a first attempt in this direction, this study also has limitations (e.g., large range of age differences, lack of a control group) that should be properly addressed and acknowledged. Thus, I'd like to recommend the following action from the authors, apart from addressing the specific comments from the reviewers:

1. E. I checked Aličković et al. (2021) and couldn't find a detailed report of the participant information. Thus, I'd suggest the authors add a table in the current manuscript (or in supplementary materials), detailing the age, gender, and hearing loss level for each of the 34 participants.

A. We added a table as Supplementary Information of the Participants section with a detailed overview of participant information. 

2. E. Age difference is indeed a concern, as hearing and auditory processing systems change dramatically over age. As the participants aged between 21 and 84 years, it is questionable whether the findings of the current study are primarily driven by a particular age group (e.g., old adults), or whether it apply universally to all ages. If the authors have had more participants, they could try to divide the participants into age groups, but that's not realistic here. I suggest the authors include age as a covariate into the statistical analyses (e.g., three-way ANOVAs) and check if age dramatically changes the results. Alternatively, if the participants are mainly older adults, perhaps the authors could conduct analysis on a subset of the data by excluding young adults, but the sample size will become smaller. Please also comment on how age difference may have impacted the results and the interpretation, as a limitation (apart from the sentence on line 500-502).

A. We agree that the age range in this experiment appears to be quite wide, going from 21 to 84 years. One issue that we clarified in the text is that the dataset included data from a 21-year-old male and a 45-year-old female, which widen the range greatly. Excluding those participants, the age range is a much more reasonable 54-84. The results did not change after excluding the two younger participants from the dataset. Please find below the details of this re-analysis (in brief, the statistical results were unaffected).

Re-analysis results

As previously done, we employed a repeated three-measures ANOVA to understand the impact of three factors: Noise Reduction (Off or On), Attention (Masker and Target), and Model (F vs. Fsh) or Model Gain (PhCat vs. PhOnset). 

For phonological encoding, our analysis revealed a significant effect of attention (F(1,31) = 23.27, p = 3.82e-05, ηp² = 0.44), with larger EEG prediction correlations for the target compared to the masker speech (rtarget > rmasker, post-hoc t-test: t(31) = 4.82, p = 3.82e-05, Cohen’s d = 0.73). There was also a main effect of model (F(1,31) = 34.39; p = 2.04e-06, ηp² = 0.53), with the F model showing greater EEG prediction correlations than its control model, Fsh, derived by re-running the TRF computation after shuffling the corresponding phoneme categories (rF > rFsh; post-hoc t-test: t(31) = 5.86, p = 2.04e-06, Cohen’s d = 0.61); and a significant Attention*Model interaction (F(1,31) = 18.23, p = 1.81e-04, ηp² = 0.38). No significant effect of Noise Reduction emerged from the ANOVA. 

For phonetic category and phoneme onset encoding, our three-way repeated-measures ANOVA revealed no significant difference between the two model gains (PhCat and PhOnset metrics) (F(1,31) = 3.47, p = 0.07, ηp² = 0.1). A significant effect of Attention also emerged (F(1,31) = 9.4, p = 0.005, ηp² = 0.24), with greater EEG prediction gains in the masker compared to the target (rmasker > rtarget, post-hoc t-test: t(31) = 3.07, p = 0.005, Cohen’s d = 0.39). Once again, there was no main effect of Noise Reduction. An interaction between attention and model gain did reach statistical significance (F(1,31) = 4.81, p = 0.037, ηp² = 0.14).

3. E. I think it'll be a good idea to acknowledge the lack of control group as another limitation of the current study.

A. We agree and have strongly highlighted that limitation at the end of the Discussion section (lines 578-580). 

4. E. In addition, I have a comment about the use of phonological processing and phonetic features.

In the abstract, the authors proposed a hypothesis of phonological encoding difficulty. However, this hypothesis was not explained further in the introduction, especially what measures would indicate phonological and non-phonological encoding, and the specific predictions about these measures. Also, the authors did not return to this hypothesis in the discussion, to discuss whether the results confirmed this hypothesis or not, and if so, whether fully or partly, etc. Throughout a large part of the manuscript, the authors used a different term "phonetic features" and seemed to use it against acoustics. I wonder what's the relationship between "phonetic features" and phonological encoding. It is true that phonology and phonetics are sometimes used interchangeably in the literature, but to many people they're distinct (e.g., phonology referring to abstract phonemes, whereas phonetics includes variations across speakers, phonetic contexts, etc.). It will be helpful if the authors could elaborate on the relationship between these terminologies if they're intended to be different, or, if they were intended to mean the same thing, then use one terminology consistently across the manuscript. I also have some issue with the conclusion that the results confirmed the hypothesis (again this was only briefly said in the abstract and not carefully discussed in the discussion). To many people, phonological encoding means abstract, acoustically invariant phonemes. However, following this definition, the results did not provide evidence for the encoding of acoustically invariant features in the target and masker speech streams. What the results showed was something very specific, the encoding of phoneme onsets. Thus, I don't think it's appropriate to simply conclude that the results confirmed the hypothesis.

A. This is a very important point. Therefore, we have revised our use of the terms “phonological encoding”, “phonetic features” and “phoneme onsets” throughout the manuscript, to ensure that they are used consistently and in a way that clearly reflects our intention. 

"Phonetic features" are used when we refer to the methodology, as we used them as features for the TRF model. By “phoneme onsets” we intend phonetic boundaries, which in practice consist of timestamps marking the start of a phonemic unit. By the neural encoding of “phonological” information we refer to the encoding of information that relates to phonemes, from phoneme onsets to acoustically-invariant responses to categories.

Reviewer #1

Thank you for taking the time to thoroughly read the manuscript.

As was already said, the topic is significant, the article is well-written, however there are still unanswered questions. Although there was a legitimate worry about the way the hidden material was presented, the study cannot be published in its current form due to significant methodological issues.

1. R. The wide age span that includes everything from young adults to the elderly. This is a crucial factor because it is well known that the ability to perceive speech declines with age, which is why the volunteers were not well chosen.

A. We agree with the Reviewer on the importance of the appropriate choice of participants depending on their age range. We verified that this is not a problem in our study. Only two participants out of thirty-four were below fifty years of age (one 21y.o. male and one 45y.o. female). Removing them from the dataset leads to a more consistent age range (54-84), without changing the result (please refer to the reply to the editor for further details on that re-analysis). 

2. R. Another significant factor influencing how speech sounds are perceived is the difference between the sexes. A quick report on the information from the hearing assessments is given. This data is essential to comprehending the characteristics of the participants and the rationale for the conclusions.

A. We have provided a detailed description of participant’s sex, age, and hearing profile, inserted as a Supplementary Table of the Participants section. 

Without concentrating on participant data, the article's emphasis is in providing a detailed presentation of the technology. Nonetheless, the article entails assessing and analysing the responses provided by the participants. which the writers mostly disregarded.

I heartily advise that a fresh investigation be conducted, but this time it should consider the research participants.

A: We share the Editor’s view on this point. We respectfully disagree with the reviewer, as we think that the article has merit despite its limitations. In fact, in addition to proposing a provocative view on how the neural processing of speech might work in HI individuals, our findings provide valuable insights that will be precious for future investigations. Considering the difficulty with the acquisition of this kind of data on this population, we think this makes our study valuable in itself. Indeed, we have done our best to improve the manuscript so that it clearly indicates its limitations and recommendations for future work in this area. 

Reviewer #3

This manuscript addresses a significant topic, employing temporal response function (TRF) analysis to explore cortical responses to phonological features within sentences among individuals with hearing impairments. It offers novel insights into the auditory attentional selection mechanisms related to hearing impairments.

1. R. The description of the results in the Abstract is somewhat unclear: “Multivariate temporal response function analyses indicated a stronger phonetic-feature encoding for target than masker speech streams. Interestingly, robust EEG encoding of phoneme onsets emerged for both target and masker streams, in contrast with previously published findings with normal hearing participants (NH) and in line with our hypothesis that speech comprehension difficulties emerge due to a robust phonological encoding of both target and masker. Finally, the neural encoding on phoneme-onsets encoding was stronger for the masker speech, pointing to a possible neural basis for the higher distractibility experienced by individuals with HI.”

These sentences introduce the results from three minor perspectives, but it's unclear whether phonetic features, phonological features, and phoneme onsets are independent. At least, I believe that phoneme onsets are a type of phonological feature, so the Abstract's depiction of the results seems confusing. In addition, it would be better to add some logical expressions instead of merely using terms like "Interestingly" and "Finally" to relate the results.

A. We agree that utilising a clear and consistent terminology is very important. We revised that issue throughout the paper. Regarding the Abstract, we restructured it to better reflect our hypothesis, ensuring that it is consistent with the terminology used in the manuscript. 

2. R. Organizing the Introduction into paragraphs would make it clearer and more logically structured for the reader.

A. We organised the Introduction into paragraphs with a clear narrative. 

3. R. “All participants were native Danish speakers and had mild-to-moderately severe symmetrical sensorineural hearing impairment, with an average 4-frequency Pure Tone Average threshold of 47.5 dB hearing level.”

It would be beneficial to provide information on the degree of hearing loss to see if there are differences among individuals with varying levels of hearing impairment. At least, authors should provide the standard deviation of 4-frequency Pure Tone Average threshold of 47.5 dB hearing level.

A. We agree with the Reviewer that providing the specific hearing thresholds for each participant would offer a more complete picture of their hearing profile. Therefore, we have added a table as Supplementary Information to the Participants section where we report single-subject hearing profiles, together with age and sex information. 

4. R. Why was only 10% of the corpus manually checked? What was the focus of this manual check, and how can you ensure the remaining 90% of the corpus is problem-free and meets the requirements?

A. In order to check the accuracy of phonetic alignment, it is usually sufficient to check the initial, middle, and final section of each audio file, in search for mismatches between the identified phonemes and the actual sounds present in the audio at that time window. If the procedure appears to be correct in these three main sections, it is likely that the intermediate portions of the audio file were also well aligned, since any problems would have resulted in a temporal shift of the detected phoneme time stamps, with alignment errors carrying over to the end of the file. So, the 10% provides a good estimate of the general performance of the forced aligner. Please note that these aligners are very precise on speech material like the ones used in this study. They can be problematic in more dynamic scenarios involving turn-taking and filler sounds (e.g., uhm), which was not our case.

5. R. How were the features shuffled (from F to Fsh), and how were the phoneme onsets extracted? Please add detailed information about signal processing.

A. Phonetic feature shuffling consisted in randomly shuffling the phonetic feature category, while keeping the time stamps constant. Such a model, where the temporal information is retained but the phonetic categorical identity is disrupted, only carries phoneme onset information, since the time of occurrence of phonemes is the only information that is left intact, but it has the same dimensionality of the unshuffled model. This methodology has been often adopted by our team and other labs in numerous previous studies (e.g., Edmund Lalor’s team, Jonathan Simon’s team, Nima Mesgarani’s). 

We have clarified the shuffling procedures and the relation to the model names throughout the Speech feature models section of the manuscript. 

6. R. What does PhOnset mean, and what is the definition of FshS?

A. We have clarified that in the text, since we agree that it could be confusing. PhOnset does not refer to a particular model of feature set, but to a contrast between two models (FshS - S).

FshS consists of temporally shuffled phonetic features (Fsh) and Spectrogram (S). The difference between the FshS model’s prediction correlations and those obtained from a model consisting of Spectrogram only, will represent the unique contribution of phoneme onsets. This prediction correlation gain, FshS - S, is called the PhOnset metric, because it quantifies the prediction gain due to phoneme onsets when accounting for the acoustics.

7. R. Figure 2c's resolution is not high enough, making the image somewhat blurry. Could you explain in detail the value of the phonetic distance feature in the article? Should the article also discuss the brain processing mechanisms of phonetic distances? Additionally, why are the points in the consonant map inconsistent between Fig 2c. and Fig S2c? What does the "?" in the consonant map represent?

A. The consonant maps in Figs 2C and S2C represent the relative distance of brain responses in response to different phonemes. This concept is based on the idea that neural responses should be more similar for similar sounds, hence the possibility of plotting a map where sounds represented with similar brain activity are plotted closer to each other. Therefore, in terms of neural p

---

## [Decision Letter · Decision Letter 2]

31 May 2024

PONE-D-23-27730R2

Cortical encoding of phonetic onsets of both attended and ignored speech in hearing impaired individuals

PLOS ONE

Dear Dr. Di Liberto,

Thank you for submitting your manuscript to PLOS ONE. After careful consideration, we have decided that your manuscript does not meet our criteria for publication and must therefore be rejected.

I am sorry that we cannot be more positive on this occasion, but hope that you appreciate the reasons for this decision.

Kind regards,

Caicai Zhang

Academic Editor

PLOS ONE

Reviewers' comments:

Reviewer's Responses to Questions

**Comments to the Author**

1. If the authors have adequately addressed your comments raised in a previous round of review and you feel that this manuscript is now acceptable for publication, you may indicate that here to bypass the “Comments to the Author” section, enter your conflict of interest statement in the “Confidential to Editor” section, and submit your "Accept" recommendation.

Reviewer #1: All comments have been addressed

Reviewer #3: All comments have been addressed

2. Is the manuscript technically sound, and do the data support the conclusions?

Reviewer #1: Partly

Reviewer #3: Yes

3. Has the statistical analysis been performed appropriately and rigorously? 

Reviewer #1: Yes

Reviewer #3: Yes

4. Have the authors made all data underlying the findings in their manuscript fully available?

Reviewer #1: No

Reviewer #3: No

5. Is the manuscript presented in an intelligible fashion and written in standard English?

Reviewer #1: Yes

Reviewer #3: Yes

6. Review Comments to the Author

Reviewer #1: Dear authors,

The authors implemented multiple revisions to the text, resulting in a notable enhancement in its overall quality. Nevertheless, there are crucial aspects of the process that have not been sufficiently explained. Below, I provide several points, including:

1) The authors included a table displaying the auditory thresholds of 21 individuals. Nevertheless, it is important to note that the referent of the values is ambiguous. The authors assert that hearing loss is bilateral, however they fail to provide independent data for each ear, which is crucial for a comprehensive tonal audiometry.

2) Additionally, they fail to provide the globally recognized rules for classifying hearing loss, such as those established by the World Health Organization (WHO) or by Lloyd and Kaplan. An unstable and inexplicable table that further consolidates concerns over the caliber of this evaluation.

3) In order to be published in a academic journal such as Plos One, the data must be easily accessible and transparent. Furthermore, it is crucial to provide the air and bone data for each ear individually.

4) Regarding the hearing loss, the approach should indicate which of the four frequencies were utilized to calculate the average.

5) The authors provide a table with 21 participants. However, they state in the paper that they eliminated people aged between 21 and 45 years old. This contradicts what is presented in the table, as there is an individual listed as 21 years old.

The faults persist and have not been rectified as anticipated following a second review, wherein the editor provided specific points of concern.

Reviewer #3: (No Response)

7. PLOS authors have the option to publish the peer review history of their article (what does this mean?). If published, this will include your full peer review and any attached files.

Reviewer #1: No

Reviewer #3: No

- - - - -

---

## [Author Response · Author response to Decision Letter 2]

20 Jun 2024

1) The authors included a table displaying the auditory thresholds of 21 individuals. Nevertheless, it is important to note that the referent of the values is ambiguous. The authors assert that hearing loss is bilateral, however they fail to provide independent data for each ear, which is crucial for a comprehensive tonal audiometry.

Authors: In our participant cohort, we only report average thresholds as no interaural asymmetry was present (> 10 dB HL difference at no more than three adjacent frequencies). That is a common way of reporting this type of data, and it is common practice also in PLOS ONE. For example, please refer to these two prior publications:

https://journals.plos.org/plosone/article?id=10.1371/journal.pone.0235782

https://journals.plos.org/plosone/article?id=10.1371/journal.pone.0213899

Given the symmetry of the thresholds, reporting a comprehensive tonal audiometry for both ears is inconsequential for our finding. We would be happy to elaborate on that in the manuscript. See the changes in lines 146-153 reflecting this information in the manuscript.

2) Additionally, they fail to provide the globally recognized rules for classifying hearing loss, such as those established by the World Health Organization (WHO) or by Lloyd and Kaplan. An unstable and inexplicable table that further consolidates concerns over the caliber of this evaluation.

Authors: All participants that participated in the data collection used for this study were hearing aid users recruited from Eriksholm’s Research Centre Hearing Aid User Database prior to the data collection. Meaning that the hearing loss classification was confirmed and known a priori. Therefore, we consider redundant the disclosure of hearing loss classification rules for this population in our manuscript. Eriksholm Research Centre abides by the globally recognized rules for classification of hearing loss, and we would be happy to elaborate on that in the manuscript. See the changes in lines 146-153 reflecting this information in the manuscript.

3) In order to be published in a academic journal such as Plos One, the data must be easily accessible and transparent. Furthermore, it is crucial to provide the air and bone data for each ear individually.

Authors: Reporting the air and bone data for each ear individual would not have any impact on the scientific relevance of our finding. Our observations are carried out within a hearing impaired cohort with specific criteria. As indicated in our response to comment 1, our procedure and reported information is in agreement with previous recent work published in PLOS ONE, so we find this comment surprising and beyond the point of this manuscript. 

Furthermore, it is within the PLOS ONE editorial policy that the data availability should be bound to local regulatory and ethical restrictions. In our view, all relevant information has been reported, guaranteeing transparency and accessibility within the ethical constraints of our data collection. Indeed, we could not share the data due to such regularity (GDPR) and ethical constraints, but that is something that other studies in PLOS ONE understands and has supported in the past, for example in this study: https://journals.plos.org/plosone/article?id=10.1371/journal.pone.0235782

However, we have provided contact details for authors to enquire about accessing the data as per the PLOS editorial policy requirement for these cases.

4) Regarding the hearing loss, the approach should indicate which of the four frequencies were utilized to calculate the average.

Authors: We agree, and we are in the position of adding that information. Specifically, the four frequencies were 500 Hz, 1000 Hz, 2000 Hz and 4000 Hz. See the changes in lines 146-153 reflecting this information in the manuscript.

5) The authors provide a table with 21 participants. However, they state in the paper that they eliminated people aged between 21 and 45 years old. This contradicts what is presented in the table, as there is an individual listed as 21 years old.

Authors: Here the reviewer has misread our reply and changes to the manuscript. We clearly stated in the revised manuscript that “Excluding those younger subjects (a 21-year-old and a 45-year-old), the age range becomes a more typical 54-84 years and, when re-running the analysis after excluding them from the dataset, the statistical results did not change”. So, as it is typical and appropriate, we re-run the analysis after excluding the participants that the reviewer indicated that could have impacted the results. As we expected, removing those participants did not change our results.

---

## [Decision Letter · Decision Letter 3]

26 Jul 2024

Cortical encoding of phonetic onsets of both attended and ignored speech in hearing impaired individuals

PONE-D-23-27730R3

Dear Dr. Di Liberto,

We’re pleased to inform you that your manuscript has been judged scientifically suitable for publication and will be formally accepted for publication once it meets all outstanding technical requirements.

Kind regards,

Andrew R Dykstra

Academic Editor

PLOS ONE

Additional Editor Comments (optional):

Reviewers' comments:

Reviewer's Responses to Questions

**Comments to the Author**

1. If the authors have adequately addressed your comments raised in a previous round of review and you feel that this manuscript is now acceptable for publication, you may indicate that here to bypass the “Comments to the Author” section, enter your conflict of interest statement in the “Confidential to Editor” section, and submit your "Accept" recommendation.

Reviewer #4: All comments have been addressed

2. Is the manuscript technically sound, and do the data support the conclusions?

Reviewer #4: Yes

3. Has the statistical analysis been performed appropriately and rigorously? 

Reviewer #4: Yes

4. Have the authors made all data underlying the findings in their manuscript fully available?

Reviewer #4: Yes

5. Is the manuscript presented in an intelligible fashion and written in standard English?

Reviewer #4: Yes

6. Review Comments to the Author

Reviewer #4: I was brought in as a third reviewer, presumably to assess the author's response to previous reviewers and provide my own review. Based on the previous responses, I agree with all of the authors' responses, and I fully endorse the publication of the article.

7. PLOS authors have the option to publish the peer review history of their article (what does this mean?). If published, this will include your full peer review and any attached files.

Reviewer #4: **Yes: **Erol Ozmeral

---

## [Editor Report · Acceptance letter]

16 Aug 2024

PONE-D-23-27730R3 

PLOS ONE

Dear Dr. Di Liberto, 

I'm pleased to inform you that your manuscript has been deemed suitable for publication in PLOS ONE. Congratulations! Your manuscript is now being handed over to our production team.

Kind regards, 

on behalf of

Dr. Andrew R Dykstra 

Academic Editor

PLOS ONE